# Identification of Policies Based on Assessment-Optimization Model to Confront Vulnerable Resources System with Large Population Scale in a Big City

**DOI:** 10.3390/ijerph182413097

**Published:** 2021-12-11

**Authors:** Xueting Zeng, Hua Xiang, Jia Liu, Yong Xue, Jinxin Zhu, Yuqian Xu

**Affiliations:** 1School of Labor Economics, Capital University of Economics and Business, Beijing 100072, China; xiangh1203@cueb.edu.cn (H.X.); 12019205067@cueb.edu.cn (Y.X.); 2Center for Population Development Research, Capital University of Economics and Business, Beijing 100072, China; 3College of Urban Economics and Public Administration, Capital University of Economics and Business, Beijing 100072, China; 4School of Geography and Planning, Sun Yat-Sen University, Guangzhou 510000, China; zhujx29@mail.sysu.edu.cn; 5Henry M. Gunn High School, Capital University of Economics and Business, 780 Arastradero Rd., Palo Alto, CA 94306, USA; yx46407@pausd.us

**Keywords:** population–resource management framework, vulnerability assessment, two-stage dynamic fuzzy programming with Hurwicz criterion, policy scenario analysis

## Abstract

The conflict between excessive population development and vulnerable resource (including water, food, and energy resources) capacity influenced by multiple uncertainties can increase the difficulty of decision making in a big city with large population scale. In this study, an adaptive population and water–food–energy (WFE) management framework (APRF) incorporating vulnerability assessment, uncertainty analysis, and systemic optimization methods is developed for optimizing the relationship between population development and WFE management (P-WFE) under combined policies. In the APRF, the vulnerability of WFE was calculated by an entropy-based driver–pressure–state–response (E-DPSR) model to reflect the exposure, sensitivity, and adaptability caused by population growth, economic development, and resource governance. Meanwhile, a scenario-based dynamic fuzzy model with Hurwicz criterion (SDFH) is proposed for not only optimizing the relationship of P-WFE with uncertain information expressed as possibility and probability distributions, but also reflecting the risk preference of policymakers with an elected manner. The developed APRF is applied to a real case study of Beijing city, which has characteristics of a large population scale and resource deficit. The results of WFE shortages and population adjustments were obtained to identify an optimized P-WEF plan under various policies, to support the adjustment of the current policy in Beijing city. Meanwhile, the results associated with resource vulnerability and benefit analysis were analyzed for improving the robustness of policy generation.

## 1. Introduction

Reliable water resources, safe gain production, and sustainable energy supply can be regarded as important factors to support urban development, which can provide basic power supply for urban operation and meet the material needs for human living [1]. However, under the background of accelerated urbanization, overpopulation growth, and rapid economic development, the demands of water–food–energy resources (WFE) are increased, which might surpass what natural or artificial systems can afford, leading to a resource crisis [2]. In particular, in some big cities such as Beijing (China) with a large population, the conflict between increasing resource demand and limited WFE carrying capacity can increase the risk of resource shortage. For instance, water shortage is a serious problem in Beijing city, where water resource per capita is one-seventh of the national average [3]. Meanwhile, the energy self-sufficiency (including coal resource, natural gas and electricity supply) of Beijing was 6% in 2017, resulting in energy supply being dependent on importation from other provinces [4]. Moreover, high urbanization under interregional coordination policy has accelerated agriculture transfer to adjacent regions (such as Hebei province), increasing its dependence level on agricultural resource importation. However, to implement water-saving and carbon emission reduction strategies in China, resource importation from other provinces may be restricted due to limited water rights and carbon emission permits. Thus, a fragile resource–supply (i.e., WFE supply) system under overpopulation growth and gathering can be deemed as an obstacle to synergistic economic gain and urban development in a big city [5]. Economic income can be obtained when the expected resource demands based on population scales are satisfied; otherwise, resource shortages generate penalties or losses. Therefore, an identification of the relationship between population development and resource management is required, attracting the attention of policymakers today. 

As early as 1968, the contradiction between population growth and resource (e.g., land resource) scarcity was disclosed in a “tragedy of the commons”, which was extended to other resources such as water, food, energy, space, and forest resources [6]. In the process of urbanization, the negative correlation between population development and natural resource supply has been verified by previous researchers [7]. Therefore, overpopulation due to urban development can raise the stress on the WFE supply system, enhancing its vulnerability due to artificial and natural driving influences [8]. Previously, various researchers paid attention to the vulnerability of resource (including WFE resources) supply capacity due to population development, whereby excessive population growth and gathering would result in resource shortage or unavailability in response to limited resource supply capacity [9,10,11,12]. For instance, Vorosmarty et al. (2000) built a hybrid numerical framework incorporating climate model outputs, water budgets, and socioeconomic information to reflect vulnerable water supply influenced by human impact, representing a potentially important facet of the larger global change question [13]. Nadine et al. (2013) used a vulnerability model to analyze two major industries on the Great Barrier Reef, reflecting the exposure level, sensitivity degree, and adaptive capacity of interaction between social components and the natural resource (including WFE) system [14]. Yang et al. (2015) developed an analytic hierarchy process combining set pair analysis model to display the vulnerability of water resources in response to population growth and climate change [15]. Chen et al. (2018) proposed a water–energy–food (WEF) condonation by PSR model, reflecting the vulnerability and coordination problems of the WEF system in northwest China from 2006 to 2015 for human survival and development. In general, previous researchers paid more attention to simulation and assessment analysis methods to reflect the relationship among population growth, economic development, and resource vulnerability [16]. In order to confront the above shortcomings, several studies associated with an adaptive management framework have been proposed. For instance, Young (2010) used resilience analysis, vulnerability assessment, and adaptation management to reflect the relationship between human lives and the ecological system, which is beneficial to prepare for brief windows of opportunity to make planned changes [17]. Li et al. (2018) combined system dynamic analysis (SD) and an optimal allocation model into a framework to support optimal water utilization according to the vulnerability of water resources, with the aim of supporting regional sustainable development in the context of population growth [18]. Xiang and Li (2020) used a functional model to assess the vulnerability of resources (including WFE) based on RAGA projection, achieving an adaptive resources management strategy to confront resource vulnerability due to excessive human activities [19].

In an adaptive management framework, the system optimization method is an effective approach to incorporate population–economy development and WFE management into a framework to coordinate various components and their relationships [20,21,22,23,24,25,26]. However, different uncertainties existing in subsystems and components of population–resource (including WFE) systems generate complex actions and reactions to other subsystems (such as population, economy, resource, and government subsystems) [27,28,29]. For example, spatiotemporal variations in natural WFE capacity caused by disparate natural features of topography and precipitation can be regarded as an important stochastic factor, leading to population fluctuation as a function of net system benefits. Meanwhile, dynamic socioeconomical development and governmental policies can result in different WFE consumption structures, which would bring about various resource stresses. Thus, a two-stage stochastic dynamic programming (TSDP) was introduced to deal with such objective or subjective randomness, which can build a link between regulated population policies and adjusted economic/resource policies under uncertainty [1]. Nevertheless, in a practical adaptive management issue, some fuzziness due to limited data acquirement (such as error estimation and lost data) can increase the difficulty of decision making, which is a challenge for TSDP [20,24]. Therefore, a credibility-based fuzzy programming (FCP) model was joined to improve the ability of tackling such precise values (e.g., vague inaccurate economic benefits or inexact losses of population adjustment) under weaker sources of information [22,25]. In addition, although random events and fuzzy data in a population–resource plan can be handled by TSDP and FCP methods, the fuzzy risk preference of policymakers would influence the robustness of the optimized population-WFE plan. Under these situations, a scenario analysis with consideration of risk preferences of policymakers based on Hurwicz criterion (SAL) can be designed to reflect the risk adaptation of policymakers in an eclectic optimistic and pessimistic manner [1,26]. However, previous studies focused little on incorporating hybrid methods (e.g., TSSP, FCP and HCA) into a framework to handle various uncertainties in a population and WFE issue.

Therefore, the objective of this study was to develop an adaptive population–WFE (P-WEF) management framework (APRF) to confront the conflict between population development and the WFE supply system. Vulnerability assessment, uncertainty analysis, and systemic optimization methods can be incorporated into this APRF as hybrid method manner to deal with multiple uncertainties, which can confront the complexity of P-WFE for policymakers. This hybrid method has the following advantages: (a) the vulnerability of WFE is calculated using an entropy-based driver–pressure–state–response (E-DPSR) model to reflect the exposure, sensitivity, and adaptability caused by population growth, economic development, and resource governance; (b) a population–resources (P-WFE) optimization can be conducted to identify various policies associated with population adjustment, resource regulation, and technique improvement; (c) a scenario-based dynamic fuzzy model with Hurwicz criterion (SDFH) is developed and embedded into P-WFE optimization to deal with uncertainties expressed as possibility and probability distributions. Meanwhile, the risk preferences of policymakers can also be reflected in an elected manner. The developed APRF was applied to a real case study of Beijing city, which has characteristics of a large population and resource deficit. The results of WFE shortages and population–economy adjustments under various scenarios were obtained to identify the optimized P-WEF under various policies. Furthermore, the obtained results associated with resource vulnerability in various scenarios and benefit analysis under various risk levels were analyzed, reflecting the tradeoff between population development and WEF management. The above results are beneficial for adjusting current policies in a risk-averse and effective manner.

## 2. Materials and Methods

### 2.1. Problem Statement

Beijing is the capital city of China with a total area of 16,412 square kilometers, located in a north temperate subhumid continental monsoon climate (located at 115.7°–117.4° E longitude, 39.4°–41.6° N latitude). It has average rainfall of 483.9 mm, which presents an uneven spatial and temporal distribution (approximately 65% of precipitation occurring in July and August; more than 77.59% of water availability distributed in Hebei province) (WRB, 2006–2017) [30]. Furthermore, although 67 types of mineral products have been found, Beijing is not a major mineral-producing area, whereby energy supply and coal products are imported from other provinces. Moreover, in the context of the coordination strategy for Beijing, Tianjin, and Hebei agglomeration (CBTH), agriculture was transferred from Beijing city to Hebei province; thus, food resources have relied on importation from circumjacent provinces in recent years. 

As a result of the speed of urban agglomeration in recent decades, the population of Beijing presented an increasing tendency from 120 to 200 billion from 2010 to 2018 [31]. Furthermore, accelerated economic growth and developed social resource support (e.g., educational resources, public service facilities, cultural deposits, employment opportunities, and other aspects) have led to constant population gathering in the core area of Beijing, which requires high-quality resource support. However, the natural resources (land, water, energy, food, forest, etc.) of Beijing struggle to meet the demands of population agglomeration due to their own characteristics (such as limited quantity, uneven spatial distribution, and interrelation of different resources). The contradiction between population development and the water–food–energy (WFE) supply requires more effective approaches to deal with these urban effects. 

Although a number of policies were introduced to relieve the WFE pressure in recent years, they brought about a number of challenges for policymakers. The polarization and siphon effects of a central city can lead to excessive population agglomeration. Excessive density of population increases the pressure on WFE in core regions [25,32]. Moreover, limited natural resources are deemed a bottleneck problem in urban development. For instance, Beijing has only one-seventh of the national average water resources per capita, which results in a serious water crisis; furthermore, other resources (such as food and mineral products) rely on importation from other provinces. In the context of the Beijing–Tianjin–Hebei coordinated development strategy, industry transformation facilitated agricultural and food product industries moving from Beijing to Heibei, reducing the self-sufficiency of food resources. Urbanization requires a high quality of energy supply (such as coal resources and corresponding derivatives), which cannot be met by an energy-deficient region dependent on importation from other provinces. However, the water-saving and carbon emission reduction strategies in China can restrict resource importation from other provinces due to limited water rights and carbon emission permits. Thus, the vulnerability of the WFE supply system has increased due to natural and artificial factors in recent decades. Although various strategies such as population adjustment based on industrial transformation, resource-saving techniques, and resource importation planning have been advocated, how to identify the interaction between population development and WFE management is a first point for remitting the resource crisis in Beijing city. Therefore, an adaptive population—WFE management framework (APRF) is required to confront the above challenges.

### 2.2. The Framework of APRF under Combined Policies

Figure 1 presents an adaptive population–WFE management framework (APRF) under combined policies to confront the conflict between rapid population development and the vulnerable WFE supply system in Beijing city. In this framework, population development (including population growth, gathering, and employment structure) and resource (including WFE) supply management can be incorporated into a system, where the population based on industrial layout (PIL) can be deemed as an important indicator to support economic growth, WFE consumption, and urban development. A proper PIL can support economic development in various industrial sectors at regional levels, which can also minimize the damage to WFE as a function of regional resource carrying capacity. Otherwise, this can lead to the vulnerability of the natural resource supply system, which may be expressed as resource shortages, economic loss, and system failure. With the agglomeration of population, the per capita possession of WFE in Beijing has decreased, and the utilization of WFE has intensified, resulting in increasing pressure on WFE vulnerability. Under these situations, the policymakers of Beijing have to adopt a series of policies and measures (such as population adjustment, industrial reallocation, technique improvement, and resource regulation) to alleviate such pressures. Nevertheless, different natural features and various governmental modes can generate a number of uncertainties and complexities. Thus, an APRF framework incorporating vulnerability assessment, population–WFE optimization, and scenario analysis can be designed to address the conflict between population development and WFE supply in Beijing city.

### 2.3. Methodology and Modeling 

#### 2.3.1. Method Development

##### Vulnerability Assessment Based on Driver–Pressure–State–Response (DPSR) Model 

In general, the vulnerability of WFE supply can be expressed as a function of exposure (EI) (the proximity of natural resources to pressures and disturbances), sensitivity (SI) (an exposed unit is affected by pressures and disturbances), and adaptability (AI) (the ability of an exposed unit to deal with and recover from adverse effects), which includes various vulnerability indices [33,34]. Since the value of VI incorporates the positive effects from EI and AI, and includes the negative effect from AI, the vulnerability index (VI) of water–food–energy resources can be defined as VI=EI+SI−AI without any empowerment [35]. Meanwhile, in order to identify the vulnerability of the WFE supply system to population scale or adjustment, an index system named the population-driven WFE vulnerability assessment framework (PWVA) was built (Table 1) [35]. Moreover, a driver–pressure–state–response (DPSR) model was introduced to reflect the cause–effect relationships and interactions between human activities and WFE supply system [36]. Following the principle of comprehensiveness, objectivity, and scientificity, the indicators in PWVA were grouped and assigned to reflect their contributions to population-driven resource vulnerability as follows: population scale, employment rate and corresponding indicators can be deemed as a drivers (denoted as “D”) to change the state of economic development and resource utilization (denoted as “S”), which would result in detriments such as water shortage, food crisis, and energy importation (denoted as “P”). Accordingly, the human response or governmental regulation (denoted as “R”) can remit the conflict between population development and the WFE supply system. The equations describing the indices based on DPSR are shown in Table 1.

Then, in order to avoid errors caused by different measured units of indicators and the mutual interference between positive and negative factors, the range method was used to standardize all indicators as follows [37]: 

For increasing class indicators,
(1a)$$x_{ij}=\frac{x_{ij}-x_{ij}}{x_{max}-x_{min}};$$

For decreasing indices,
(1b)$$x_{ij}=\frac{x_{max}-x_{ij}}{x_{max}-x_{min}}.$$ Here, $x_{ij}$ is the j indicator data of the *i* year, *x_max_* is the maximum value of the j indicator, and xmin is the minimum value of the *j* indicator. Moreover, the entropy value method is introduced as the weight analysis tool, which has the advantages of simple calculation, original data utilization, less information loss, and reduced subjective information [38,39]. The entropy weight calculation formula can be expressed as follows:(1c)$$e_{j}=-k\sum_{i=1}^{m}y_{ij}lny_{ij},$$
(1d)yij=xij∑i=1mxij,
(1e)k=1lnmxij,
(1f)wj=1−ej∑j=1n(1−ej),
where $e_{j}$ is the information entropy of each indicator, *m* is the number of years, *n* is the number of indicator items, and $x_{ij}$ is the standardized value of the *j* indicator in *t* year. 

According to the definition of VI (in Table 1) and entropy weight calculation method (Equations (1a)–(1f)), the exponent of the domain layer and target layer can be calculated by weighted summation as follows: (1g)$$P_{r-vi}=\sum_{n=1}^{m}x_{r-ij}z_{j},$$
where $P_{r-vi}$ represents the r index under the domain layer and the target layer, $x_{r-ij}$ is the specific index (the standardized value of the j index in t year) under the domain layer or the criterion layer, and $z_{j}$ is the weight of this index calculated according to the entropy value method. 

In the survey, since the data used for comprehensive evaluation were all standardized data, this would increase the operative errors. However, standardized data fully retain the information of the original data, which can also reflect the change rate of indicators to a certain extent [40]. Under these situations, the variation trend of different resource data in Beijing results in the intercept term being eliminated after standardization; however, some data still have a time trend. Furthermore, the AIC (Akaike information criterion) was used to determine the optimal lag order for the augmented Dickey–Fuller unit root test (ADF) [41]. Thus, the stationarity of standardized data was analyzed in Table 2. It can be seen that the ADF statistical values of all index data in Beijing were all less than the 1% critical value, revealing a stable trend. 

##### Optimization Based on a Scenario-Based Dynamic Fuzzy Model with Hurwicz Criterion (SDFH) Method

Taking into account the vulnerability assessment of WFE supply system, the optimization method should be considered to balance the relationship between population development and WFE management. However, random events (such as random rainfall, sudden agricultural reduction, and dynamic energy plans) impacting the constrained reliability would influence the accuracy of calculation. Therefore, a two-stage stochastic dynamic programming (TSDP) was introduced to build a link between expected targets (i.e., expected resource (WEF) demand based on population scale) and random events (i.e., the reduction in WEF supply capacity caused by drought, plant disease, and energy restriction) as follows [20]:(2a)$$\mathrm{Max}\;f^{}=uw-\sum_{h=1}^{r}p_{h}^{}q(v,\delta_{h}^{}),$$
subject to
(2b)$$R(\delta_{h}^{})w+S(\delta_{h}^{})v=g(\delta_{h}^{}),\delta_{h}^{}\in \Omega ,$$
(2c)$$uw-\sum_{h=1}^{r}q(v,\delta_{h}^{})\leq w,$$
(2d)aw≤c,
(2e)w≥0,
(2f)v≥0.

In Model (2a), if the expected WEF demand (initial target or first-stage variable) based on the population scale can be satisfied, the first-stage benefit (i.e., *uw*) would be obtained; otherwise, a loss or recourse action (i.e., *q*(*ν*, *δ_h_*)) would be generated. This means that the first benefit can be rectified by the second penalty (e.g., $\sum_{h=1}^{r}p_{h}^{}q(v,\delta_{h}^{})$) when a random event occurs; *P_h_* is the probability of a random event. According to the vulnerability of the WFE supply system driven by population, the possibility of WFE shortage would rectify the expected WEF, as shown in Model (2c) [1,20]. However, data uncertainties due to limited data acquirement (such as error estimation and lost data) cannot be handled by TSDP with probabilistic distributions [25,42,43,44]. Therefore, a fuzzy credibility constrained programming (FCP) was combined with TSDP as follows: (3a)$$Cr\{aw\leq \tilde{c}\}\geq \alpha .$$

On the basis of the concept of fuzzy credibility, the credibility measure (*Cr*) can be expressed as $Cr\left\{\varsigma \leq s\right\}=\frac{1}{2}(Pos\left\{\varsigma \leq s\right\}+Nec\left\{\varsigma \leq s\right\})$ [22]. In general, the credibility level should be greater than 0.5. Thus, Model (3a) can be proven as follows:(3b)$$Cr\left\{\tilde{\varsigma }\geq s\right\}\geq \alpha \Leftrightarrow s\leq (2-2\alpha )\varsigma_{2}+(2\alpha -1)\varsigma_{1}\Leftrightarrow s\leq \varsigma_{2}+(1-2\alpha )(\varsigma_{2}-\varsigma_{1}).$$

In a practical a two-stage stochastic fuzzy credibility programming (TFC) issue, the input of left- or right-hand variables are impacted by various factors, thereby requiring a scenario analysis (SA) [45]. Thus, various adaptive scenarios can be designed to balance the relationship between expected target and actual WFE supply. However, the risk attitudes of policymakers can impact the generation of scenarios due to the experiences and personality traits of policymakers, which cannot be handled by TFC. Therefore, a scenario analysis with Hurwicz criterion was introduced to reflect the vague risk adaptation of policymakers, which is beneficial to obtain compromising alternatives based on eclectic optimistic and pessimistic criteria [46,47,48]. The scenario-based dynamic fuzzy model with Hurwicz criterion (SDFH) method can be expressed as follows:(4a)$$\mathrm{Max}\;f_{Hurwicz}^{}=\{\beta *U_{}^{opt}+(1-\beta )*U_{}^{pec}\},$$
subject to
(4b)$$Cr\left\{[uw-\sum_{h=1}^{r}p_{h}^{}q(v,\delta_{h}^{})]\leq U_{}^{opt}\right\}\geq \eta_{},\;i=1,2,\ldots ,I,$$
(4c)$$Cr\left\{[uw-\sum_{h=1}^{r}p_{h}^{}q(v,\delta_{h}^{})]\geq U_{}^{pec}\right\}\geq \eta_{},\;i=1,2,\ldots ,I,$$
(4d)$$R(\delta_{h}^{})w+S(\delta_{h}^{})v=g(\delta_{h}^{}),\;\delta_{h}^{}\in \Omega ,$$
(4e)$$uw-\sum_{h=1}^{r}q(v,\delta_{h}^{})\leq w,$$
(4f)$$aw\leq c_{n}^{2}+(1-2\alpha )(c_{n}^{2}-c_{n}^{1}),$$
(4g)w≥0,
(4h)v≥0.

#### 2.3.2. Modeling Formulation for Practical Application

In a practical framework of APRF, the initial population policy is pre-regulated on the basis of urban planning, which can support economic development, but is restricted by WFE carrying capacity (or natural supply capacity). In general, the initial population policy is calculated as a function of the current population situation (with existing techniques and resource regulation) as a baseline scenario. When the expected demand of WFE based on initial population policy or scale can be satisfied by WEF, a first-stage benefit is generated. However, since the WFE carrying capacity can be influenced by some artificial or objective factors (such as random rainfall or resource regulation change), the shortage of WFE based on the initial population scale can generate an economic loss, which can be regarded as a second-stage recourse action. Under these situations, policymakers would implement a series of policies (including industrial structure adjustment and corresponding employed population change), which can be deemed further second-stage recourse actions to remit the losses of WFE shortage. In a practical APRE, population scale in various industries can be seen as an indicator (i.e., decision variable) to consume WFE resources, which can generate variation in employment structure, industrial layout, and resource use pattern, leading to different WFE stresses. Thus, policymakers can make policy adjustments (such as population adjustment in different industries, resource regulations in WEF supply plans, and technique improvement in supply capacity) to reduce WFE shortages, which can maximize system benefit in a risk-averse and effective manner as follows:(5a)$$\begin{array}{ll} \mathrm{Max}\;f_{Hurwicz} & =\{\beta *U_{}^{opt}+(1-\beta )*U_{}^{pec}\}\\ & =\{\beta *{{>[\mathrm{ICR}}_{t}^{}+{\mathrm{BLA}}_{t}^{}+{\mathrm{BCT}}_{t}^{}]}_{}^{opt}+(1-\beta )*{{[\mathrm{ICR}}_{t}^{}+{\mathrm{BLA}}_{t}^{}+{\mathrm{BCT}}_{t}^{}]}_{}^{pec}\} \end{array}.$$

The notations of objective functions, decision variables, and parameters are shown in the Abbreviations. In Model (5a), *f_Hurwicz_* is the total system benefit with the Hurwicz criterion, which includes the income from the current population situation and the corresponding loss from WFE shortage (${\mathrm{ICR}}_{t}^{}$), the benefit and loss from the WFE supply capacity based on population adjustment (${\mathrm{BLA}}_{t}^{}$), and the benefit and loss from the WFE supply capacity based on population adjustment (${\mathrm{BLA}}_{t}^{}$) with consideration of the risk adoption based on the Hurwicz criterion as follows:(1)Income from current population situation and corresponding loss from WFE shortage (${\mathrm{ICR}}_{t}^{}$):(5b)$$\begin{array}{l} (\sum_{t=1}^{3}\sum_{l=1}^{2}IMP_{tl}^{}*\tilde{B}MP_{tl}^{}-\sum_{l=1}^{2}\sum_{t=1}^{3}\sum_{h=1}^{3}p_{th}^{}*RMP_{tlh}^{}*\tilde{L}MS_{tl}^{})+(\sum_{t=1}^{3}\sum_{j=1}^{1}IAP_{tj}^{}*\tilde{B}AP_{tj}^{}-\\ \sum_{j=1}^{1}\sum_{t=1}^{3}\sum_{h=1}^{3}p_{th}^{}*RAP_{tjh}^{}*\tilde{L}AS_{tj}^{})+(\sum_{k=1}^{4}\sum_{t=1}^{3}IEP_{tk}^{}*\tilde{B}IP_{tk}^{}-\sum_{k=1}^{4}\sum_{t=1}^{3}\sum_{h=1}^{3}p_{th}^{}*REP_{tkh}^{}*\tilde{L}IS_{tk}^{})\\ +(\sum_{g=1}^{3}\sum_{t=1}^{3}ISP_{tg}^{}*\tilde{B}SP_{tg}^{}-\sum_{g=1}^{3}\sum_{t=1}^{3}\sum_{h=1}^{3}p_{th}^{}*RSP_{tgh}^{}*\tilde{L}SS_{tg}^{}) \end{array}.$$

In Model (5b), three industrial (including agriculture, industry, and service sectors) and one municipal sectors in three periods (t denoted as various periods) are considered. Since WFE consumption patterns vary in different industrial and municipal sectors, two plants (*l* = 1 represents resources for urban residents; *l* = 2 represents resources for rural residents) in the municipal sector, one plant (*j* = 1 denotes irrigation in agriculture) in the agriculture sector, four plants (*k* = 1 and 2 represent high- and medium-consumption industrial plants, *k* = 3 represent other industrial plants, and *k* = 4 represent energy-supply plants) in the industrial sector, and three plants (*g* = 1 represents the traditional service industry; *g* = 2 represents other service industrial plants; *g* = 3 represents green service industrial plants) in the service sector are shown in Model (5b). In Model (5b), the expected population scales (including living population and employed population according to the current industrial structure) regarded as first-stage variables (i.e., $IMP_{tl}^{}$, $IAP_{tj}^{}$, $IEP_{tk}^{}$, $ISP_{tg}^{}$) can bring about incomes or benefits (i.e., $BMP_{tl}^{}$, $BAP_{tj}^{}$, $BIP_{tk}^{}$, $BSP_{tg}^{}$), when the resource (including water, coal, and food resources) supply capacity can satisfy or support such population scales. The economic data associated with net system benefit for the population in various sectors ($BMP_{tl}^{}$, $BAP_{tj}^{}$, $BIP_{tk}^{}$, $BSP_{tg}^{}$) are provided in Table 3. If the WFE resources cannot meet the expected demand based on the population scale, resource-deficient losses are generated, where $RMP_{tlh}^{}$, $RAP_{tjh}^{}$, $REP_{tkh}^{}$, and $RSP_{tgh}^{}$ can be deemed as second-stage variables.
(2)Benefit and loss from WFE supply capacity based on population adjustment (${\mathrm{BLA}}_{t}^{}$):(5c)$$\begin{array}{l} \left\{\sum_{l=1}^{2}\sum_{t=1}^{3}IMP_{lt}^{}*\tilde{B}MP_{lt}^{}*[wre_{lt}^{}*(1-\eta_{lt}^{})+cre_{lt}^{}*(1-\delta_{lt}^{})]-\sum_{j=1}^{1}\sum_{i=1}^{3}\sum_{h=1}^{3}p_{th}^{}*AMP_{lth}^{}*\tilde{L}MP_{lt}^{}*\;[wre_{lt}^{}*(1-\eta_{lt}^{}\right)\\ +cre_{lt}^{}*(1-\delta_{lt}^{})]\}\{\sum_{j=1}^{1}\sum_{t=1}^{3}IAP_{tj}^{}*\tilde{B}AP_{tj}^{}*[wre_{tj}^{}*(1-\eta_{tj}^{})+cre_{tj}^{}*(1-\delta_{tj}^{})]-\sum_{j=1}^{1}\sum_{i=1}^{3}\sum_{h=1}^{3}p_{th}^{}*AAP_{tjh}^{}*\tilde{L}AP_{tj}^{}*\\ \;[wre_{tj}^{}*(1-\eta_{tj}^{})+cre_{tj}^{}*(1-\delta_{tj}^{})]\}+\{[\sum_{k=1}^{4}\sum_{t=1}^{3}IEP_{tk}^{}*\tilde{B}IP_{tk}^{}*[wre_{tk}^{}*(1-\eta_{tk}^{})+cre_{tk}^{}*(1-\delta_{tk}^{})]-\\ \sum_{k=1}^{4}\sum_{i=1}^{3}\sum_{h=1}^{3}p_{th}^{}*AEP_{tkh}^{}*\tilde{L}IP_{tk}^{}*[wre_{tk}^{}*(1-\eta_{tk}^{})+cre_{tk}^{}*(1-\delta_{tk}^{})]\}+[\sum_{g=1}^{3}\sum_{t=1}^{3}ISP_{tg}^{}*\tilde{B}S_{tg}^{}*\;[wre_{tg}^{}*(1-\eta_{tg}^{})\\ +cre_{tg}^{}*(1-\delta_{tg}^{})]-\sum_{g=1}^{3}\sum_{i=1}^{3}\sum_{h=1}^{3}p_{th}^{}*ASP_{tgh}^{}*\tilde{L}SP_{tg}^{}*[wre_{tg}^{}*(1-\eta_{tg}^{})+cre_{tg}^{}*(1-\delta_{tg}^{})]\} \end{array}.$$

The initial population policy and corresponding population scale can support economic development, albeit while consuming various resources, leading to WFE stresses or losses. Thus, the policies associated with population adjustment (i.e., $AMP_{tlh}^{}$, $AAP_{tjh}^{}$, $AEP_{tkh}^{}$, $ASP_{tgh}^{}$) would remit the losses of WFE shortage (i.e., $LMS_{tl}^{}$, $LAS_{tj}^{}$, $LIS_{tk}^{}$, $LSS_{tg}^{}$), but would generate penalties (i.e., $LMP_{tl}^{}$, $LAP_{tj}^{}$, $LIP_{tk}^{}$, $LSP_{tg}^{}$) due to the shrinking of the economy. The corresponding economic data are provided in Table 4. Furthermore, the policies associated with technique improvement (e.g., the coefficient of water resource consumption or coal use and the coefficient of water-saving or coal-saving) were considered as policy scenarios to remit WFE pressure, as shown in Table 3.
(3)Benefit and cost from technique improvement (${\mathrm{BCT}}_{t}^{}$):(5d)$$\begin{array}{l} \left\{\sum_{l=1}^{2}\sum_{t=1}^{3}IMP_{lt}^{}*\tilde{L}MS_{lt}^{}*[ee_{lt}^{}*(1-\phi_{lt}^{})+ae_{lt}^{}*(1-\mu_{lt}^{})]-\sum_{j=1}^{1}\sum_{i=1}^{3}*RMP_{lth}^{}*CMS_{lt}^{}*\;[ee_{lt}^{}*(1-\phi_{lt}^{}\right)\\ +ae_{lt}^{}*(1-\mu_{lt}^{})]\}+\{\sum_{j=1}^{1}\sum_{t=1}^{3}IAP_{tj}^{}*\tilde{L}AP_{tj}^{}*[ee_{tj}^{}*(1-\phi_{tj}^{})+ae_{tj}^{}*(1-\mu_{tj}^{})]-\sum_{j=1}^{1}\sum_{i=1}^{3}*RAP_{tjh}^{}*CAS_{tj}^{}*\\ \;[ee_{tj}^{}*(1-\phi_{tj}^{})+ae_{tj}^{}*(1-\mu_{tj}^{})]\}+\{[\sum_{k=1}^{4}\sum_{t=1}^{3}IEP_{tk}^{}*\tilde{L}IP_{tk}^{}*[ee_{tk}^{}*(1-\phi_{tk}^{})+ae_{tk}^{}*(1-\mu_{tk}^{})]-\\ \sum_{k=1}^{4}\sum_{i=1}^{3}*REP_{tkh}^{}*CIS_{tk}^{}*[ee_{tk}^{}*(1-\phi_{tk}^{})+ae_{tk}^{}*(1-\mu_{tk}^{})]\}+[\sum_{g=1}^{3}\sum_{t=1}^{3}ISP_{tg}^{}*\tilde{L}S_{tg}^{}*\;[ee_{tg}^{}*(1-\phi_{tg}^{})\\ +ae_{tg}^{}*(1-\mu_{tg}^{})]-\sum_{g=1}^{3}\sum_{i=1}^{3}*RSP_{tgh}^{}*CSS_{tg}^{}*[ee_{tg}^{}*(1-\phi_{tg}^{})+ae_{tg}^{}*(1-\mu_{tg}^{})]\} \end{array}.$$


In Model (5d), the expected population can discharge pollutants, which would disturb the WFE supply capacity to an extent. Therefore, various recycling techniques were considered in the scenario analysis, including the coefficient of recycling by technique improvement, as shown in Table 3. Here, η and δ are the improvement ratios of the resource-saving technique (%), while μ and ϕ are the improvement ratios of the retreatment technique (%). However, the costs of technique improvement (i.e., $CMS_{lt}^{}$, $CAS_{tj}^{}$, $CIS_{tk}^{}$, $CSS_{tg}^{}$) should be considered.

Moreover, various constraints associated with available resources (WFE), population development, and other economic development scales under various scenarios can be considered as follows:
(1)Constraints of available water resources and corresponding resource regulation:(6a)$$Cr\{\sum_{h=1}^{3}\sum_{t=1}^{3}\tilde{V}w_{ht}^{}=\sum_{h=1}^{3}\sum_{t=1}^{3}[({\tilde{R}}_{ht}^{}-{\tilde{H}}_{ht}^{}-{\tilde{G}}_{ht}^{})\}\geq \alpha .$$(6b)$$\begin{array}{l} Cr\{[wre_{lt}^{}*(1-\eta_{lt}^{})(\sum_{l=1}^{2}\sum_{t=1}^{3}IMP_{lt}^{}-\sum_{j=1}^{1}\sum_{i=1}^{3}\sum_{h=1}^{3}p_{th}^{}*RMP_{lth}^{})]+[wre_{tj}^{}*(1-\eta_{tj}^{})(\sum_{j=1}^{1}\sum_{t=1}^{3}IAP_{tj}^{}-\\ \sum_{j=1}^{1}\sum_{i=1}^{3}\sum_{h=1}^{3}p_{th}^{}*RAP_{tjh}^{})]+[wre_{tj}^{}*(1-\eta_{tj}^{})*(\sum_{k=1}^{4}\sum_{t=1}^{3}IEP_{tk}^{}-\sum_{k=1}^{4}\sum_{i=1}^{3}\sum_{h=1}^{3}p_{th}^{}*REP_{tkh}^{})]+\\ \left[wre_{tg}^{}*(1-\eta_{tg}^{})*(\sum_{g=1}^{3}\sum_{t=1}^{3}ISP_{tg}^{}-\sum_{g=1}^{3}\sum_{i=1}^{3}\sum_{h=1}^{3}p_{th}^{}*RSP_{tgh}^{}]\leq (1-\xi_{w}^{})*\tilde{V}w_{ht}^{}\}\geq \alpha \right\} \end{array}.$$



Model (6a) shows the constraint of available water resources based on current water resource load, which is equal to the total water availability (including surface and underground water resources) (${\tilde{R}}_{hi}^{}$) minus loss of water (${\tilde{H}}_{hi}^{}$) (including evaporation/infiltration from river) and water requirements of watercourse (${\tilde{G}}_{hi}^{}$). Model (6b) displays that water shortage would occur according to expected demands and available water resources ($\tilde{V}w_{hj}^{}$), which can be expressed in a credibility-based fuzzy manner due to imprecise information. However, in order to save water, policymakers design resource limit targets (resource regulation scenarios) in planning periods, where $\xi_{w}^{}$ is the reduced ratio of the resource limit for the resource saving target (%); corresponding policy scenarios associated with resource regulation are displayed in Table 3.
(2)Constraints of available coal resources and corresponding resource regulation:(6c)$$\begin{array}{l} Cr\{[cre_{lt}^{}*(1-\delta_{lt}^{})(\sum_{l=1}^{2}\sum_{t=1}^{3}IMP_{lt}^{}-\sum_{j=1}^{1}\sum_{i=1}^{3}\sum_{h=1}^{3}p_{th}^{}*RMP_{lth}^{})]+[cre_{tj}^{}*(1-\delta_{tj}^{})(\sum_{j=1}^{1}\sum_{t=1}^{3}IAP_{tj}^{}-\\ \sum_{j=1}^{1}\sum_{i=1}^{3}\sum_{h=1}^{3}p_{th}^{}*RAP_{tjh}^{})]+[cre_{tj}^{}*(1-\delta_{tj}^{})*(\sum_{k=1}^{4}\sum_{t=1}^{3}IEP_{tk}^{}-\sum_{k=1}^{4}\sum_{i=1}^{3}\sum_{h=1}^{3}p_{th}^{}*REP_{tkh}^{})]+\\ \left[cre_{tg}^{}*(1-\delta_{tg}^{})*(\sum_{g=1}^{3}\sum_{t=1}^{3}ISP_{tg}^{}-\sum_{g=1}^{3}\sum_{i=1}^{3}\sum_{h=1}^{3}p_{th}^{}*RSP_{tgh}^{}]\leq (1-\xi_{c}^{})*\tilde{V}c_{ht}^{}\}\geq \alpha \right\} \end{array}.$$(3)Constraints of available food resources and corresponding resource regulation:(6d)$$\begin{array}{l} Cr\{[[wre_{tj}^{}*(1-\eta_{tj}^{})(\sum_{j=1}^{1}\sum_{t=1}^{3}IAP_{tj}^{}-\sum_{j=1}^{1}\sum_{i=1}^{3}\sum_{h=1}^{3}p_{th}^{}*RAP_{tjh}^{})]{\;*\mathrm{Fre}}_{tj}^{}+[wre_{tg}^{}*(1-\eta_{tg}^{})*(\sum_{g=1}^{3}\sum_{t=1}^{3}ISP_{tg}^{}\\ -\sum_{g=1}^{3}\sum_{i=1}^{3}\sum_{h=1}^{3}p_{th}^{}*RSP_{tgh}^{}]{*\mathrm{Fre}}_{tg}^{}-(\sum_{l=1}^{2}\sum_{t=1}^{3}IMP_{lt}^{}-\sum_{j=1}^{1}\sum_{i=1}^{3}\sum_{h=1}^{3}p_{th}^{}*RMP_{lth}^{})]{*\mathrm{Fre}}_{lt}^{}+\leq (1-\xi_{f}^{})*\tilde{V}f_{ht}^{}\}\geq \alpha \} \end{array}.$$

Models (6c) and (6d) present the recourse actions of coal and food shortages to expected demands, which can be restricted by limited available coal and food resources ($\tilde{V}c_{ht}^{}$ and $\tilde{V}f_{ht}^{}$) and corresponding resource regulation ($\xi_{c}^{}$, $\xi_{f}^{}$). In this constraint, policy analysis associated with resource regulation is considered, as shown in Table 3.


(4)Constraints of living population scale for agricultural sector:(6e)$$IAL_{tj}^{\min}\leq \sum_{j=1}^{1}\sum_{t=1}^{3}IAP_{tj}^{}\leq IAL_{tj}^{\max}.$$(5)Constraints of employed population scale for industrial sector:(6f)$$IIL_{tk}^{\min}\leq +\sum_{k=1}^{4}\sum_{t=1}^{3}IEP_{tk}^{}\leq IIL_{tk}^{\max}.$$(6)Constraints of employed population scale for service sector:(6g)$$ISL_{tg}^{\min}\leq \sum_{g=1}^{3}\sum_{t=1}^{3}ISP_{tg}^{}\leq ISL_{tg}^{\max}.$$


Models (6e) to (6g) present the scales of agriculture, industry, service, and population development in Beijing city. Here, $IAL_{tj}^{\max}$, $IIL_{tk}^{\min}$, $IIL_{tk}^{\max}$, $ISL_{tg}^{\min}$, and $ISL_{tg}^{\max}$ are the minimum and maximum population scales for agricultural, industrial, and service sectors (person). 


(7)Constraints of capacity of water-saving and energy efficiency techniques:(6h)$$\begin{array}{l} \{[\sum_{l=1}^{2}\sum_{t=1}^{3}IMP_{lt}^{}*ee_{lt}^{}*(1-\phi_{lt}^{})-\sum_{j=1}^{1}\sum_{i=1}^{3}\sum_{h=1}^{3}p_{th}^{}*RMP_{lth}^{}*\;ee_{lt}^{}*(1-\phi_{lt}^{})]+\{[\sum_{j=1}^{1}\sum_{t=1}^{3}IAP_{tj}^{}*ee_{tj}^{}*(1-\\ \phi_{tj}^{})]-\sum_{j=1}^{1}\sum_{i=1}^{3}\sum_{h=1}^{3}p_{th}^{}*RAP_{tjh}^{}*\;ee_{tj}^{}*(1-\phi_{tj}^{})]+[\sum_{k=1}^{4}\sum_{t=1}^{3}IEP_{tk}^{}*ee_{tk}^{}*(1-\phi_{tk}^{})-[\sum_{k=1}^{4}\sum_{i=1}^{3}\sum_{h=1}^{3}p_{th}^{}*REP_{tkh}^{}*ee_{tk}^{}\\ {}*(1-\phi_{tk}^{})]+[\sum_{g=1}^{3}\sum_{t=1}^{3}ISP_{tg}^{}*\;ee_{tg}^{}*(1-\phi_{tg}^{})-\sum_{g=1}^{3}\sum_{i=1}^{3}\sum_{h=1}^{3}p_{th}^{}*RSP_{tgh}^{}*ee_{tg}^{}*(1-\phi_{tg}^{})]\}\leq \tilde{C}w_{ht}^{}\}\geq \alpha \end{array}.$$
(6i)$$\begin{array}{l} \left\{\sum_{l=1}^{2}\sum_{t=1}^{3}[IMP_{lt}^{}*ae_{lt}^{}*(1-\mu_{lt}^{})-\sum_{j=1}^{1}\sum_{i=1}^{3}\sum_{h=1}^{3}p_{th}^{}*RMP_{lth}^{}*ae_{lt}^{}*(1-\mu_{lt}^{})]+[\sum_{j=1}^{1}\sum_{t=1}^{3}IAP_{tj}^{}*+ae_{tj}^{}*(1-\mu_{tj}^{}\right)\\ -\sum_{j=1}^{1}\sum_{i=1}^{3}\sum_{h=1}^{3}p_{th}^{}*RAP_{tjh}^{}*\;ae_{tj}^{}*(1-\mu_{tj}^{})]+[\sum_{k=1}^{4}\sum_{t=1}^{3}IEP_{tk}^{}*ae_{tk}^{}*(1-\mu_{tk}^{})-[\sum_{k=1}^{4}\sum_{i=1}^{3}\sum_{h=1}^{3}p_{th}^{}*REP_{tkh}^{}*ae_{tk}^{}*(1\\ -\mu_{tk}^{})]+[\sum_{g=1}^{3}\sum_{t=1}^{3}ISP_{tg}^{}*\;ae_{tg}^{}*(1-\mu_{tg}^{})-\sum_{g=1}^{3}\sum_{i=1}^{3}\sum_{h=1}^{3}p_{th}^{}*RSP_{tgh}^{}*ae_{tg}^{}*(1-\mu_{tg}^{})]\}\leq \tilde{C}c_{ht}^{}\}\geq \alpha \end{array}.$$


Models (6h) and (6i) demonstrate the capacity of the saving technique and recycling technology. Here, $Cw_{ht}^{}$ and $Cc_{ht}^{}$ are the maximal capacity of technique improvement.


(8)Constraints of Hurwicz criterion:(6j)$$Cr\{{(\mathrm{ICR}}_{t}^{}+{\mathrm{BLA}}_{t}^{}+{\mathrm{BCT}}_{t}^{})\leq U_{}^{opt}\}\geq \alpha .$$
(6k)$$Cr\{{(\mathrm{ICR}}_{t}^{}+{\mathrm{BLA}}_{t}^{}+{\mathrm{BCT}}_{t}^{})\geq U_{}^{pec}\}\geq \alpha .$$(9)Constraints of economic benefit and loss: (6l)$$LAP_{tjh}^{}\leq BAP_{tj}^{},LIP_{tkh}^{}\leq BEP_{tk}^{},LSP_{tgh}^{}\leq BSP_{tg}^{}.$$(10)Non-negative constraints: (6m)$$LAP_{tjh}^{},BAP_{tj}^{},\;LIP_{tkh}^{},BEP_{tk}^{},LSP_{tgh}^{},BSP_{tg}^{}\geq 0.$$
(6n)$$RAP_{tjh}^{},BAP_{tj}^{},\;RIP_{tkh}^{},REP_{tk}^{},RSP_{tgh}^{},RSP_{tg}^{}\geq 0.$$


Models (6j) to (6n) present the Hurwicz criterion, as well as various economic benefit or loss and non-negative restrictions.

### 2.4. Data Acquisition

Table 4 shows the economic data expressed as fuzzy values in three planning periods (5 years for one planning period), which were estimated using the statistical yearbooks of Beijing from 2000 to 2018 with consideration of the economic growth rate [23,24]. Moreover, the random variation in WEF supply capacity could lead to seasonal/uneven distribution of resource availability. For instance, spatial and temporal variation in water availability can result from an uneven distribution of precipitation, which was calculated using previous simulation studies on the annual rainfall of Beijing city (2000 to 2017) [25]. Thus, the probability of low, medium, and high levels of water resources was obtained as 0.2, 0.6, and 0.2 [49,50]. In the study region, the main food production from agriculture can be influenced by rainfall, while the energy supply from hydropower stations can be affected by precipitation under a stable thermal power supply. Thus, the probability of low, medium, and high levels of food and energy resources would be same as the water resources. Furthermore, since the available resources can be influenced by artificial factors (e.g., data deficits and estimation errors), fuzzy programming with a credibility measure was used for the expression of fuzziness. 

Table 3 displays the combined policy scenarios of Beijing city, with the aim of reflecting the policy tradeoff among population, technique improvement, and resource regulation. In this study, scenario 0 (S0) represents the basic policy scenario based on the current population—WFE situation, which can generate optimal results according to existent population conditions, economic development, technique levels, and resource regulation. On the basis of S0, three policy scenarios associated with technique improvement (S1 to S3) were designed for three planning periods, using the “empirical method” and “expert consultation method”. For instance, the lowest elevated values of resource-use efficiency and retreatment ratio (i.e., 5%) were calculated using previous values (i.e., “empirical method”). With consideration of the speed of technological development in recent years, the highest elevated values of resource-use efficiency and retreatment ratio were estimated by experts (i.e., “expert consultation method”), i.e., 25%. According to the lowest and highest elevated values of resource-use efficiency and retreatment ratio, the medium value (15%) was obtained by the “expert consultation method”. Then, two scenarios (S4 and S5) associated with resource regulations were designed according to the same principle, where two levels (5% and 15%) of lower resource limitation were considered. In addition, two policy scenarios (S6 and S7) involving combined policies took technique improvement and resource regulation into account to reflect the tradeoff among various policies.

## 3. Results and Discussion

### 3.1. WFE Vulnerability under Basic Policy Scenario (S0) 

Figure 2 shows the vulnerability of WFE from 2000 to 2017. The results present that the vulnerability of WFE increased then decreased, with the inflection points occurring in 2012, 2014, and 2016 in line with strategies for population adjustment and technique improvement in Beijing city. In the comparison of exposure (EI), sensitivity (SI), and adaptability (AI) levels, the results reveal that SI levels would be higher than EI levels, demonstrating that population adjustment would influence the WFE system (indicating that WFE system is sensitive to population adjustment). However, the low levels of AI suggest that the effectiveness of government responses to WFE stresses would be relatively low, particularly in 2011. 

In general, according to the increase in WFE vulnerability under S0, the failure of the WEF supply system would increase, which would result in a resource shortage. Figure 3 displays the WFE shortages among various sectors under S0 (α = 0.6 and β = 0.99) as follows: (a) water shortages would be mainly influenced by rainfall, presenting a higher value in the dry season and vice versa; (b) the highest coal shortages would occur in industrial plants, particularly in heavy-consumption and other industrial plants (denoted as “HID” and “OID”), representing the main energy-use sectors in Beijing city; (c) food shortages would mainly occur in residential areas due to the large population scale and agriculture transformation to Hebei, which is also influenced by rainfall levels; (d) lower credibility satisfaction levels (α-level) would result in higher resource deficits and vice versa. 

In order to remit resource deficits, the adjustment of population is analyzed in Figure 4, which displays population adjustments among different industrial plants based on water–coal optimization under S0 when α = 0.99 and β = 0.6. The results show that the highest resource-deficient sectors (e.g., HIDs and OIDs) would not lead to the highest population reduction (population adjustment based on industrial transformation), since they had higher incomes or benefits despite great resource stress. On the contrary, the highest population reduction would occur in other service industrial plants (denoted as “OSEs”), due to their lowest benefit per population. Furthermore, the WFE deficit of HIDs and OIDs can be remitted by technique improvement due to their higher-yielding supports; thus, OSEs (deemed as a lower-yielding sector) would adjust the more employed population to relieve WFE pressures and vulnerability.

### 3.2. WFE Shortage and Population Adjustment under Various Policy Scenarios (S1 to S7)

Figure 5 displays the solutions for population adjustments and corresponding WFE shortages in different sectors under various policy scenarios associated with technique improvement (S1 to S3), when α = 0.99 and β = 0.6. The results show that an improvement of resource utilization efficiencies would lead to lower WFE shortages instead of higher population adjustments. Thus, a higher technique improvement level would generate a lower population adjustment. However, technique improvement would generally require financial support; thus, the tradeoff between the income from technique improvement and the cost of technique improvement should be considered.

Figure 6 presents the satisfaction of WFE targets and corresponding population adjustments under various policy scenarios associated with governmental regulation (S4 to S5), when α = 0.99 and β is 0.6. The results display that the adjustment of population would be reduced by contractible resource regulation. In the comparison of various resources deficits in different sectors, the lowest water satisfaction would take place in the agricultural sector according to lower water resource limits, while the lowest coal satisfaction would occur in heavy-consumption industrial plants and energy-supply plants (e.g., HIDs and ESIs). Although resource regulation can reduce water satisfaction in some plants by reducing direct economic income in the short term, this would compel the companies to save resources to lessen the losses of WFE deficit by pursuing technique improvement, which is more beneficial to technical progress and the water-saving goal achievement in the long term.

Figure 7 shows the lower population reduction based on combined policies (S1 to S7) compared to S0 (α = 0.6 and β = 0.99). The obtained results demonstrate that an adaptive policy associated with technique improvement can lessen the population reduction in various industrial sectors, whereas a policy associated with resource regulation would increase the population reduction. For example, the lowest population reduction would occur in S3 (the highest technique improvement), while the highest would take place in S5. The combined policies would generate a comprehensive population reduction (such as S6 and S7). The policy tradeoff among population adjustment, resource regulation, and technique improvement can relieve the contradiction between population development and WFE supply capacity.

### 3.3. System Benefit and Vulnerability Analysis under Various Policy Scenarios (S0 to S7) 

Figure 8 presents the vulnerability of resources under S0 to S7, reflecting the variation trend of exposure–sensitivity–adaptability under S0 to S7. With increasing intensity of population adjustment, the exposure of natural resources would decrease accordingly. A conservation population regulation policy would reduce the exposure by about 5.4%; in contrast, an aggressive scenario would reduce the exposure by 30.8%. This indicates that population adjustment, technological progress, and government regulation of resources would not have significant impacts on the exposure level. The adaptability of natural resources in Beijing is influenced by both technological progress and government regulation policies, among which government regulation policies have a greater impact on the adaptability.

Figure 9 presents the system benefits and corresponding risk levels under various policy scenarios (from S0 to S7) with the Hurwicz criterion. The obtained results demonstrate the following: (a) the adjustment of population can remit the resource deficit to reduced loss of shortage, which is beneficial for a higher system benefit; (b) although policy scenarios associated with technique improvement can relieve resource stresses, they require financial support, which would generate promotion expenses, leading to decreased benefits (as shown in S1 to S3); (c) policy scenarios associated with resource regulations (e.g., S4 to S5) would generate a reduction in the employed population, leading to lower system benefits; (d) the combined policy scenarios would generate comprehensive benefits (e.g., S6 and S7); (e) a higher α level corresponding to a higher credibility measure (lower violated risk) would generate a lower benefit and vice versa; (f) as the scenario assumption is influenced by the risk preference of policymakers, the Hurwicz criterion (reflected in the β-level) was considered, showing that a higher β level corresponding to a more optimistic attitude can generate a higher benefit, but result in a higher violated risk in the process of decision making and vice versa. Increases in technique requirement and governmental regulation are not suitable for sustainable development of population and WFE management in Beijing. Thus, in general, a combined optimistic–pessimistic result can be obtained when the β level is 0.5.

### 3.4. Discussion

In this study, population development (such as population scale, growth speed, gathering, and *employment structure*) can be deemed as an important indicator to support economic development and resource consumption, which are pre-regulated at the beginning of planning periods by policymakers for urban planning. Under these situations, population development can be regarded as a driving factor in the vulnerability assessment of WFE (as shown in Table 1), as well as a decision variable in systemic optimization of the P-WEF issue in Beijing city. Population scales and the employed population in various industrial sectors can influence the resource demand and consumption patterns (various consumption coefficients), which would result in differences in WFE vulnerability, resource shortage, optimal population adjustment, and system benefits. 

In a practical P-WEF optimization issue, the implementation of resource regulation and technique improvement can remit resource shortages and vulnerabilities, which would weaken the role of population adjustment. However, a rational design scheme for the initial population policy of the planning period can be considered an effective approach to reduce the losses/penalties of future population adjustment from the long-term perspective of urban development. Thus, various initial population policy scenarios were designed according to the historical situation of population change in Beijing (from 2000 to 2017) and the “14th Five Year Plan” of Beijing as follows: (a) the high and low population development scenarios (S8 and S9) were assumed considering 2.8% and 0.11% growth rates of the population size; (b) three scenarios (S10 to S12) associated with the adjustment of employed population scales were designed according to industrial information and reformed in the “14th Five Year Plan”, whereby 1.6%, 4.6%, and 7.1% reductions in the employed population in high-resource-consumption industries (such as agricultural and industrial sectors) were considered. Then, the vulnerability of resources (WEF) under S0 and S8 to S12 were obtained as shown in Figure 10. The results show that both a high growth mode (S8) and a low growth mode (S9) of population size would increase the WFE vulnerability in the short term taking into consideration the current population. Furthermore, it is shown that industrial transformation in Beijing can shrink the employed population scale in high-resource0consumption industries, which would reduce the corresponding WEF vulnerability. 

According to the above analysis, the objective of this study was achieved, whereby an adaptive population–resource (including WFE) management framework (APRF) incorporating vulnerability assessment, uncertainty analysis, and systemic optimization methods was developed to optimize the relationship between population development and the water–food–energy (WFE) supply system under combined policies. According to the application of the developed APRF in Beijing, subobjectives were addressed. The vulnerability of resource (including WFE) supply driven by population was calculated and analyzed using the entropy-based driver–pressure–state–response (E-DPSR) model, reflecting the existing WEF vulnerability expressed as exposure, sensitivity, and adaptability based on historical data in a big city. According to the vulnerability assessment, a population–resource (P-WFE) optimization analysis was conducted to identify various policies associated with population adjustment, resource regulation, and technique improvement. A scenario-based dynamic fuzzy model with Hurwicz criterion (SDFH) was developed and embedded into the population–resource (P-WFE) optimization analysis to deal with various types of uncertainties. SDFH can not only build a link between predefined population scale/expected WFE demand and resource shortage penalties due to random water/energy flow, but also handle fuzziness due to data deficiencies. In addition, it is effective in reflecting the risk preferences of policymakers in the process of decision making. Various results associated with WFE shortages, population–economy adjustments, resource vulnerability, and system benefits under various policy scenarios were analyzed, reflecting the tradeoff between population development and WFE management in a risk-averse manner. The above-obtained objectives can facilitate the adjustment of population–resource policies in a big city.

## 4. Conclusions

With the aid of practical implications of the developed APRF framework in Beijing city, a number of discoveries were made. Firstly, the current population scale and WEF demand cannot accommodate to the regional resource supply capacity in Beijing city, resulting in resource shortages. Secondly, although industrial transformation can prompt the adjustment of the employed population structure to reduce WFE stress and vulnerability, excessive and irrational population policies would damage the smooth operation of the economy in a big city. Thirdly, low efficiencies in the current WFE use pattern and technique level of Beijing enhance the vulnerability of the resource supply system. Although policies associated with technique improvement for resource saving are effective in addressing the conflict among population, WFE, and economy, a higher investment of technology would hinder their application. Thus, how to make a comprehensive policy for balancing the relationships among population adjustment, resource regulation, and technique improvement taking into account the risk preferences of policymaker is an important issue for sustainable development of Beijing. Fourthly, individual polices (such as improvement of resource utilization efficiency, resource regulation, and employed population adjustment based on industrial transformation) have their own advantages in terms of resource shortage reduction, but there are limitations of the high cost of generalization and direct income reduction in the short term. Thus, how to balance the tradeoff between benefit and cost in the long term can be challenging for regional policymakers. Lastly, the differences in the risk attitude of policymakers when confronting uncertain information (e.g., fuzzy resource supply capacity, dynamic expected target, and risk preference) would generate varied policies, which would influence the P-WEF strategy.

Therefore, specific recommendations are proposed. Regional resource carrying capacity should be deemed as an important impact factor in a rational population policy, which could reduce the losses of resources deficits in a big city with a large population scale. Increasing the entry standards for high-consumption enterprises can allow adjusting the employed population structure, which is beneficial for reducing WFE stress/vulnerability and ensuring high-quality economic development. Furthermore, a greener and cleaner production mode should be encouraged to generate a new population employment situation, remitting resource stress from the consumption side. A combination of policies such as market admittance, governmental support, and financial subsidies to support technique improvement should be carried out, which can stimulate resource saving and recycling. Moreover, the government should promote the concept of resource saving to improve the deficiencies of resource consumption from the consumer’s perspective. The tradeoff between economic benefits and costs of various policy scenarios should be designed not only in the short term, but also in the long term, which can maximize the positive effects, while minimizing risks to a great extent. Lastly, the risk preferences of policymaker should be considered in comprehensive governance strategies or policies to fortify the robustness of population–resource optimization, thereby achieving sustainable development in Beijing city. 

## Figures and Tables

**Figure 1 ijerph-18-13097-f001:**
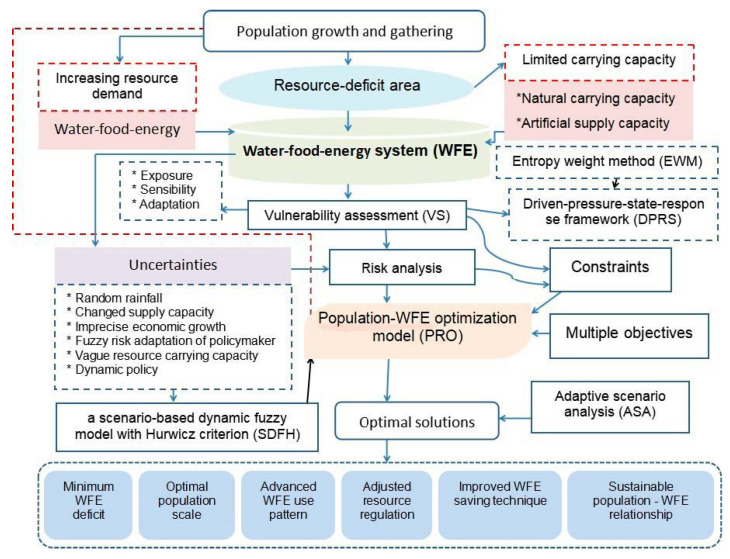
An adaptive population—resources management framework (APRF) under combined policies and its application in Beijing.

**Figure 2 ijerph-18-13097-f002:**
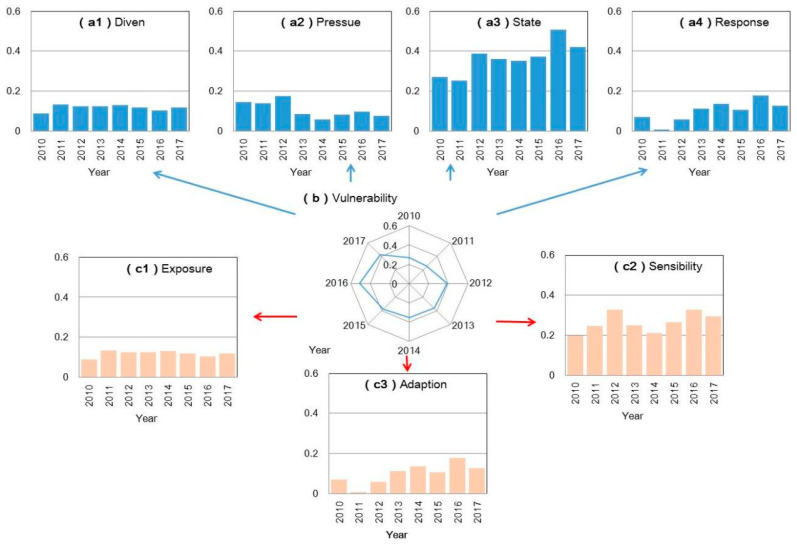
The vulnerability of resources (WEF) from 2000 to 2017 under S0 (without policy adjustment). ((**a1**–**a4**) are the measure of vulnerability based on Driver–Pressure–State–Response (DPSR) Model; (**b**) is the total vulnerability level; (**c1**–**c3**) are the exposure, sensitivity, and adaptability levels in vulnerability).

**Figure 3 ijerph-18-13097-f003:**
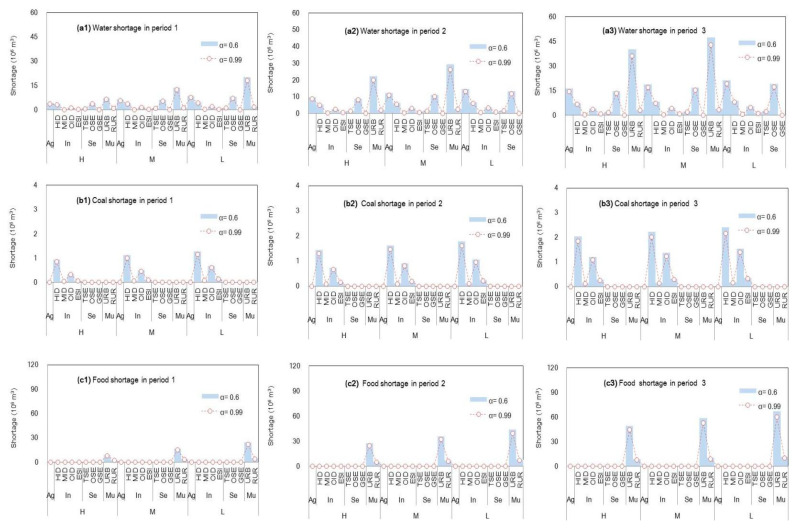
Total water–food–energy shortages among various sectors under S0 (α = 0.6 and β = 0.99) (agricultural sector denoted as “Ag”, industrial sector denoted as “In”, service sector denoted as “Se”, municipal sector denoted as “Mu”; heavy-consumption industry plant denoted as “HID”, medium-consumption industry plant denoted as “MID”, other industry plant denoted as “OID”, energy-supply industry plant denoted as “ESI”, traditional service industry plant denoted as “TSE”, other service industry plant denoted as “OSE”, environmental protection industry plant denoted as “GSE”, urban municipal consumption plant denoted as “URB”, rural consumption plant denoted as “RUR”; high level of resource availability denoted as “H”, medium level of resource availability denoted as “M”, low level of resource availability denoted as “L”) ((**a1**–**a3**) are water shortages in various periods; (**b1**–**b3**) are the coal shortages in various periods; (**c1**–**c3**) are the food shortages in various periods).

**Figure 4 ijerph-18-13097-f004:**
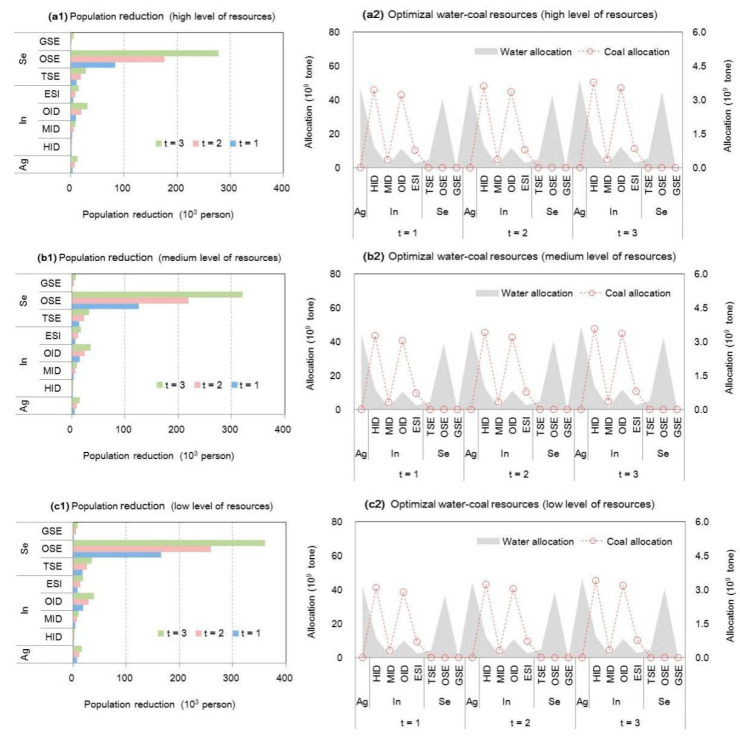
Population adjustments among different industrial plants based on water–coal optimization under S0 when α = 0.99 and β = 0.6 (agricultural sector denoted as “Ag”, industrial sector denoted as “In”, service sector denoted as “Se”, municipal sector denoted as “Mu”; heavy-consumption industry plant denoted as “HID”, medium-consumption industry plant denoted as “MID”, other industry plant denoted as “OID”, energy-supply industry plant denoted as “ESI”, traditional service industry plant denoted as “TSE”, other service industry plant denoted as “OSE”, environmental protection industry plant denoted as “GSE”, urban municipal consumption plant denoted as “URB”, rural consumption plant denoted as “RUR”) ((**a1**,**a2**) are population adjustment and corresponding optimized water-coal resources under high level of resources; (**b1**,**b2**) are population adjustment and corresponding optimized water-coal resources under medium level of resources; (**c1**,**c2**) are population adjustment and corresponding optimized water-coal resources under low level of resources).

**Figure 5 ijerph-18-13097-f005:**
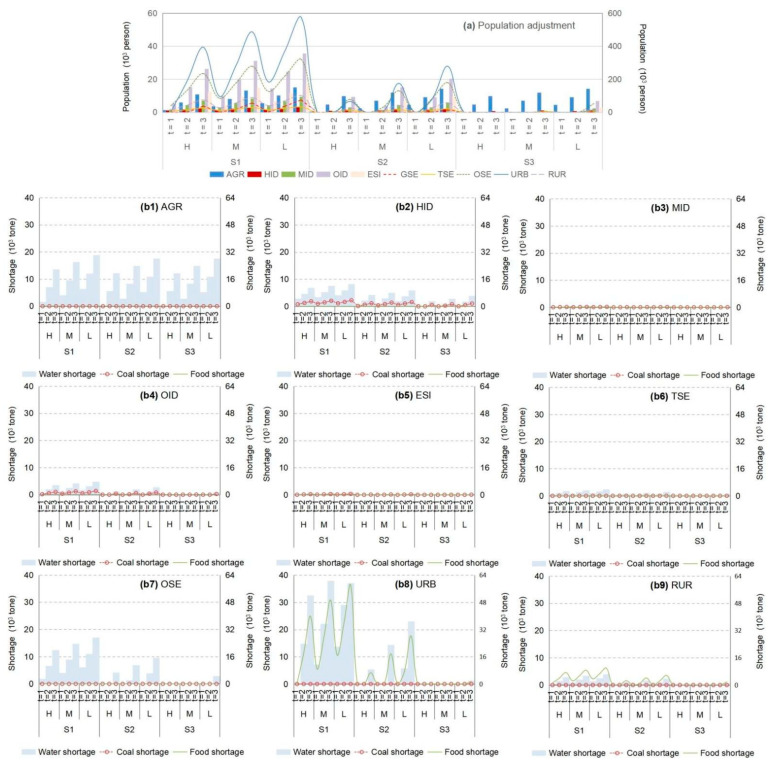
Population adjustments and corresponding water–food–coal shortages in different sectors under S1 to S3 when α = 0.99 and β = 0.6 (agricultural sector denoted as “Ag”, industrial sector denoted as “In”, service sector denoted as “Se”, municipal sector denoted as “Mu”; heavy-consumption industry plant denoted as “HID”, medium-consumption industry plant denoted as “MID”, other industry plant denoted as “OID”, energy-supply industry plant denoted as “ESI”, traditional service industry plant denoted as “TSE”, other service industry plant denoted as “OSE”, environmental protection industry plant denoted as “GSE”, urban municipal consumption plant denoted as “URB”, rural consumption plant denoted as “RUR”; high level of resource availability denoted as “H”, medium level of resource availability denoted as “M”, low level of resource availability denoted as “L”) ((**a**) is total population adjustment under S1, S2 and S3; (**b1**–**b9**) are water–food–coal shortages in different sectors under S1 to S3).

**Figure 6 ijerph-18-13097-f006:**
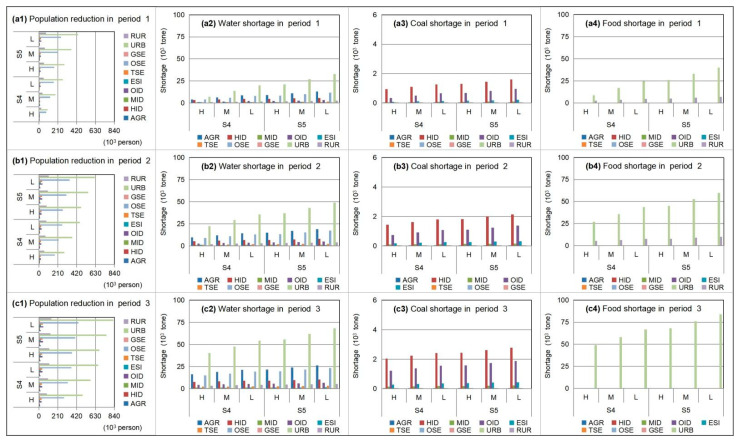
The satisfactions of water–food–energy target and corresponding population adjustments under S4 to S5 when α = 0.99 and β = 0.6 (agricultural sector denoted as “Ag”, industrial sector denoted as “In”, service sector denoted as “Se”, municipal sector denoted as “Mu”; heavy-consumption industry plant denoted as “HID”, medium-consumption industry plant denoted as “MID”, other industry plant denoted as “OID”, energy-supply industry plant denoted as “ESI”, traditional service industry plant denoted as “TSE”, other service industry plant denoted as “OSE”, environmental protection industry plant denoted as “GSE”, urban municipal consumption plant denoted as “URB”, rural consumption plant denoted as “RUR”; high level of resource availability denoted as “H”, medium level of resource availability denoted as “M”, low level of resource availability denoted as “L”) ((**a1**–**a4**) are population adjustment and corresponding WEF shortages under S4 and S5 in period 1; (**b1**–**b4**) are population adjustment and corresponding WEF shortages under S4 and S5 in period 2; (**c1**–**c4**) are population adjustment and corresponding WEF shortages under S4 and S5 in period 3).

**Figure 7 ijerph-18-13097-f007:**
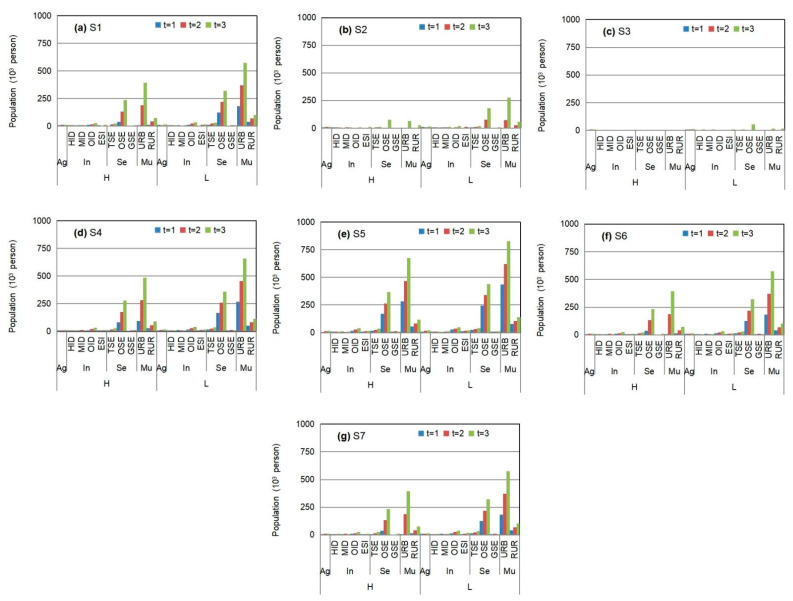
The lower population reduction based on combined policies (S1 to S7) compared to S0 (α = 0.6 and β = 0.99) (agricultural sector denoted as “Ag”, industrial sector denoted as “In”, service sector denoted as “Se”, municipal sector denoted as “Mu”; heavy-consumption industry plant denoted as “HID”, medium-consumption industry plant denoted as “MID”, other industry plant denoted as “OID”, energy-supply industry plant denoted as “ESI”, traditional service industry plant denoted as “TSE”, other service industry plant denoted as “OSE”, environmental protection industry plant denoted as “GSE”, urban municipal consumption plant denoted as “URB”, rural consumption plant denoted as “RUR”; high level of resource availability denoted as “H”, low level of resource availability denoted as “L”) ((**a**–**g**) are the population reduction under S1 to S7).

**Figure 8 ijerph-18-13097-f008:**
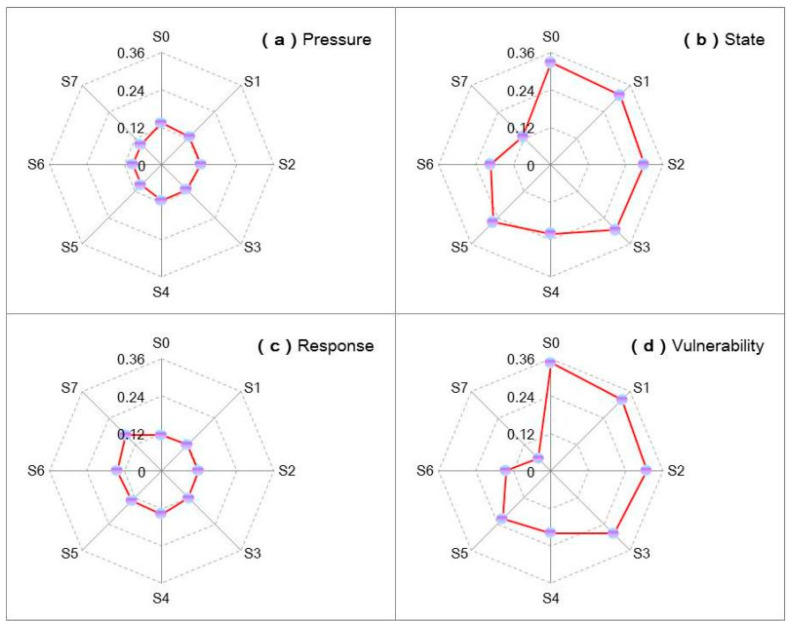
The vulnerability of resources (WEF) under S0 to S7. ((**a**–**c**) are the measure of vulnerability based on Driver–Pressure–State–Response (DPSR) Model; (**d**) is the total vulnerability level).

**Figure 9 ijerph-18-13097-f009:**
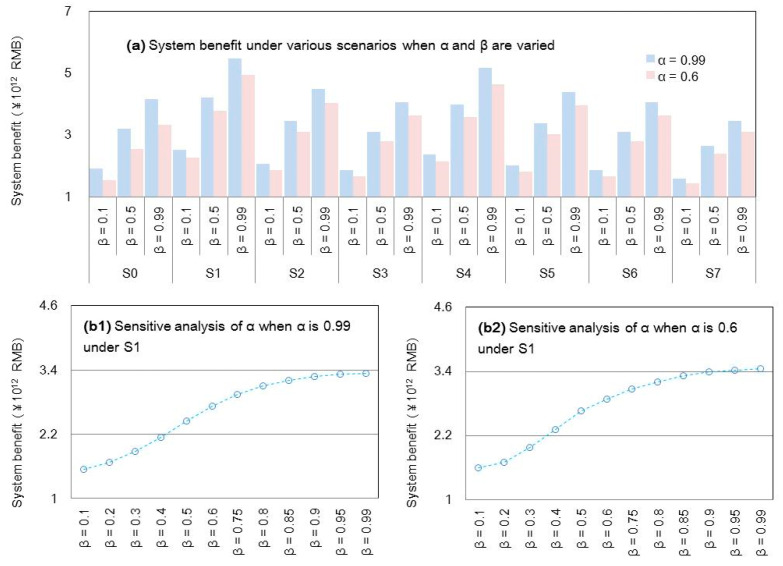
System benefit and risk analysis under S0 to S7.

**Figure 10 ijerph-18-13097-f010:**
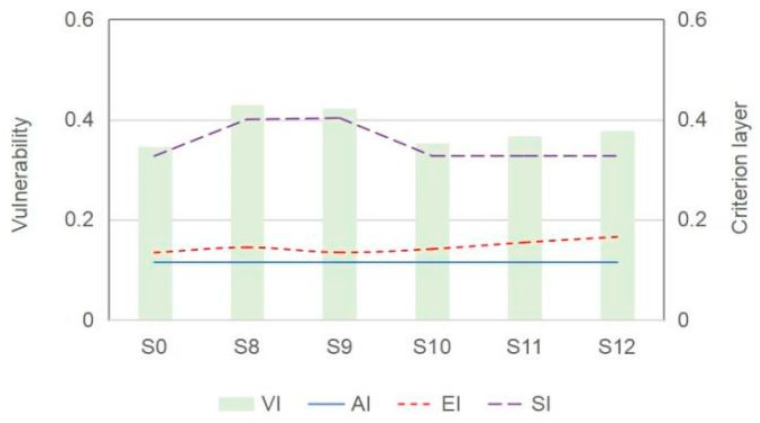
The vulnerability of resources (WEF) under S0 and S8 to S12.

**Table 1 ijerph-18-13097-t001:** Population-driven WFE vulnerability assessment framework.

DPSR Framework	Index	Equation	Vulnerability
Driver	Population scale	Direct data index	Exposure (EI)
	Employment rate	Employment/total labor force	
	The proportion of population for agriculture in industries	Agricultural population/population	
	The proportion of population for industry in industries	Industrial population/population	
	The proportion of population for service industry in industries	Service population/population	
	R&D people	Direct data index	
Pressure	The proportion of agriculture in industries	R&D people in agriculture industry/R&D people	
	The proportion of industry in industries	R&D people in industry/R&D people	
	The proportion of service industry in industries	R&D people in service industry/R&D people	
	Land utilization rate	The area of land developed/total land area	
	Energy yield-to-consumption ratio	Energy output/energy consumption	
	The efficiency of energy utilization	Industrial GDP/energy consumption	
	Total water availability	Direct data index	
	Forest coverage rate	Forest area/total land area	
State	Water resources per capita	Total water resources/population	Sensitivity (SI)
	Forest area per capita	Total forest area/population	
	Energy self-sufficiency gap	Energy consumption − energy output	
	Water shortage rate	(Water consumption − water resource)/water resource	
	Per capital greening gap	(Per capital green area − standard green value)/standard green value	
	The rate of food self-sufficiency	Grain consumption/grain output	
Response	Ecological environment investment index	The government energy conservation/general budget	Adaptability (AI)
	The government energy conservation	Direct data index	
	Sewage treatment rate	Amount of sewage purification/total sewage	
	Water-saving percentage	Circulating water consumption/water consumption	
	Intensity of soil erosion control	Water and soil loss after treatment/water and soil loss before treatment × 100%	

**Table 2 ijerph-18-13097-t002:** Consistency test.

Variable	Test Type (C, T, P)	ADF Statistic	1% Threshold	5% Threshold	10% Threshold	Conclusion
Population scale	(0, 1, 1)	−3.645	−3.75	−3	−2.63	Stable performance
Employment structure	(0, 0, 1)	−3.019	−3.75	−3	−2.63	Stable performance
The proportion of population for agriculture in industries	(0, 1, 2)	−3.104	−3.75	−3	−2.63	Stable performance
The proportion of population for industry in industries	(0, 1, 1)	−3.933	−3.75	−3	−2.63	Stable performance
The proportion of population for service industry in industries	(0, 1, 1)	−3.306	−3.75	−3	−2.63	Stable performance
R&D people	(0, 1, 1)	−4.355	−3.75	−3	−2.63	Stable performance
The proportion of agriculture in industries	(0, 1, 1)	−3.572	−3.75	−3	−2.63	Stable performance
The proportion of industry in industries	(0, 1, 2)	−3.962	−3.75	−3	−2.63	Stable performance
The proportion of service industry in industries	(0, 1, 2)	−3.451	−3.75	−3	−2.63	Stable performance
Land utilization rate	(0, 0, 1)	−3.971	−3.75	−3	−2.63	Stable performance
Energy yield-to-consumption ratio	(0, 0, 1)	−4.946	−3.75	−3	−2.63	Stable performance
The efficiency of energy utilization	(0, 1, 2)	−3.199	−3.75	−3	−2.63	Stable performance
Total water availability	(0, 0, 1)	−4.895	−3.75	−3	−2.63	Stable performance
Forest coverage rate	(0, 1, 0)	−4.619	−3.75	−3	−2.63	Stable performance
Water resources per capita	(0, 0, 2)	−5.544	−3.75	−3	−2.63	Stable performance
Forest area per capita	(0, 1, 0)	−5.388	−3.75	−3	−2.63	Stable performance
Water shortage rate	(0, 0, 2)	−4.514	−3.75	−3	−2.63	Stable performance
Per capita greening gap	(0, 1, 2)	−6.062	−3.75	−3	−2.63	Stable performance
Ecological environment investment index	(0, 1, 1)	−4.927	−3.75	−3	−2.63	Stable performance
The government energy conservation	(0, 0, 1)	−4.273	−3.75	−3	−2.63	Stable performance
Sewage treatment rate	(0, 1, 1)	−3.852	−3.75	−3	−2.63	Stable performance
Intensity of soil erosion control	(0, 0, 2)	−5.984	−3.75	−3	−2.63	Stable performance

**Table 3 ijerph-18-13097-t003:** Policy scenario.

Scenario	Assumption				
	Improvement of Technique Efficiency		Lessen the Limit of Resource Based on Resource Saving		
	Resource Use Efficiency	Retreatment Ratio	Water Resources	Coal Resources	Food Supply
S0	0%	0%	0%	0%	0%
S1	5%	5%	0%	0%	0%
S2	15%	15%	0%	0%	0%
S3	25%	25%	0%	0%	0%
S4	0%	0%	5%	5%	5%
S5	0%	0%	15%	15%	15%
S6	5%	5%	5%	5%	5%
S7	15%	15%	15%	15%	15%

**Table 4 ijerph-18-13097-t004:** Economic data.

		Period 1	Period 2	Period 3
Net System Benefit for Population in Various Sectors (10^3^ RMB/Person)				
Municipal sectors	Urban human living	(0.96, 1.02, 1.06)	(0.99, 1.04, 1.08)	(1.02, 1.06, 1.12)
	Rural human living	(0.42, 0.47, 0.50)	(0.45, 0.49, 0.51)	(0.47, 0.51, 0.55)
Agricultural sector	Food resource supply	(2.20, 2.63, 2.88)	(2.32, 2.68, 2.96)	(1.93, 2.36, 2.68)
Industrial sector	Heavy resource-consumption plants	(192.46, 202.32, 206.32)	(178.23, 183.76, 194.13)	(152.36, 148.32, 132.23)
	Medium resource-consumption plants	(12.58, 13.36, 14.58)	(10.82, 11.08, 12.98)	(9.98, 8.25, 7.26)
	Other industrial plants	(29.82, 31.15, 32.87)	(27.32, 29.55, 31.64)	(26.01, 28.25, 29.08)
	Energy-supply plants	(6.21, 7.05, 8.02)	(6.87, 7.21, 8.98)	(7.17, 8.83, 9.76)
Service sector	Traditional service plants	(18.13, 21.43, 25.98)	(20.23, 28.32, 40.12)	(34.32, 45.49, 66.32)
	Other service plants	(21.32, 28.60, 32.32)	(27.32, 32.29, 40.87)	(30.82, 38.41, 47.32)
	Environmentally friendly service plants	(22.32, 26.63, 28.89)	(29.86, 34.81, 38.86)	(36.87, 42.00, 48.82)
Net loss for various sectors (10^3^ RMB/person)				
Municipal sectors	Urban human living	(1.90, 1.96, 2.02)	(1.93, 1.99, 2.06)	(1.97, 2.01, 2.08)
	Rural human living	(0.80, 0.85, 0.90)	(0.85, 0.89, 0.93)	(0.89, 0.93, 0.96)
Agricultural sector	Food resource supply	(2.86, 3.18, 3.98)	(2.92, 3.23, 4.02)	(2.76, 2.95, 3.12)
Industrial sector	Heavy resource-consumption plants	(192.46, 202.32, 206.32)	(178.23, 183.76, 194.13)	(152.36, 148.32, 132.23)
	Medium resource-consumption plants	(12.58, 13.36, 14.58)	(10.82, 11.08, 12.98)	(9.98, 8.25, 7.26)
	Other industrial plants	(29.82, 31.15, 32.87)	(27.32, 29.55, 31.64)	(26.01, 28.25, 29.08)
	Energy-supply plants	(7.42, 8.46, 9.02)	(7.82, 8.97, 9.92)	(14.76, 15.40, 16.89)
Service sector	Traditional service plant	(22.32, 23.32, 24.56)	(32.32, 39.24, 42.12)	(40.32, 46.56, 49.12)
	Other service plants	(26.12, 27.12, 28.89)	(30.21, 36.94, 43.01)	(36.01, 41.60, 46.12)
	Environmentally friendly service plants	(27.32, 28.92, 31.21)	(32.72, 38.99, 41.34)	(39.32, 41.60, 44.34)

## Data Availability

The data presented in this study are available on request from the corresponding author.

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
