# Peer review of "Identification of Policies Based on Assessment-Optimization Model to Confront Vulnerable Resources System with Large Population Scale in a Big City"

_ijerph, 2021, doi:10.3390/ijerph182413097_

Round 1

Reviewer 1 Report

Brief summary

This paper is interesting to those who deal with city management, particularly population growth and water-food-energy management.

The focus of the paper is an adaptive population-resources management framework developed to optimize population growth and water-food-energy management under combined policies. The city of Beijing (China) is presented as a case study, from 2000 to 2017, considering three periods (although the readers don’t know the start and the end of each period) and seven policy scenarios.

The paper follows a traditional structure, and it has a considerable set of references.

However, the English writing is not perfect, and the explanation of the methodology and the presentation of the results are not quite explicit and easy to understand.

Although I think it is an interesting paper, in my opinion authors should be encouraged to rewrite the paper, perhaps with the help of a native English speaker, taking into consideration the comments presented above and below, and resubmit it.

Specific comments

Line 159: “from 12 to 200 billion from 2010 to 2018”. Is this correct?

Figure 2: is it 0.99 or 0.6?

Figure 3: what is the meaning of “when is 0.6”?

Figure 4: is it 0.99 or 0.6?

Figure 5: is it 0.99 or 0.6?

Figure 6: is it 0.99 or 0.6?

Figure 7: is it 0.99 or 0.6? What is the meaning of “when is 0.6”?

Figure 8: is it 0.99 or 0.6?

Some of the graphs are not easy to read.

Some graphs have too large vertical scales when compared to the plotted values.

In the graphs, it would be interesting to know what is H, M, L, Ag, In, Se, Mu, HID, MID, OID, ES, …, TSE, OSE and GSE. Some of these are explained in the text but most of them not.

Author Response

Nov 28th, 2021

Dear editor,

Reference Number: ijerph-1455435R1

Manuscript Title: Identification of policies based on asessment-optimzation model to confront vulnerable resources system with large population scale in a big city

Thank you very much for your email message of Nov 18th, 2021. We are glad to learn that the above-referenced paper can be accepted for publication after being revised based on the reviewers’ comments. We much appreciate your and reviewers' insightful comments and suggestions, and have revised the paper accordingly. Attached please find:

(1) revised manuscript, and

(2) responses to the reviewers’ comments

We are very grateful for the kindness and carefulness of editor. We are deeply grateful to editor for his/her careful review, which can be expressed in the “Acknowledgment” section of the revised manuscript.

Thank you very much again for your time and kind consideration. We look forward to your response.

Sincerely,

X.T. Zeng, Ph.D.

J. Liu, Ph.D.

Reviewer 2 Report

Please find the comments in the attached file

Author Response

Nov 28th, 2021

Dear editor,

Reference Number: ijerph-1455435R1

Manuscript Title: Identification of policies based on assessment-optimization model to confront vulnerable resources system with large population scale in a big city

Thank you very much for your email message of Nov 18th, 2021. We are glad to learn that the above-referenced paper can be accepted for publication after being revised based on the reviewers’ comments. We much appreciate your and reviewers' insightful comments and suggestions, and have revised the paper accordingly. Attached please find:

(1) revised manuscript, and

(2) responses to the reviewers’ comments

We are very grateful for the kindness and carefulness of editor. We are deeply grateful to editor for his/her careful review, which can be expressed in the “Acknowledgment” section of the revised manuscript.

Thank you very much again for your time and kind consideration. We look forward to your response.

Sincerely,

X.T. Zeng, Ph.D.

J. Liu, Ph.D.

Round 2

Reviewer 2 Report

Please find the comments in the attached file

Author Response

Dear reviewer,

We much appreciate the your insightful comments and suggestions, and have revised the paper accordingly. Please find the attachment!

Thank you very much again for your time and kind consideration. We look forward to your response.

Sincerely,

X.T. Zeng, Ph.D.

J. Liu, Ph.D.
